# Silver-Based Surface Plasmon Sensors: Fabrication and Applications

**DOI:** 10.3390/ijms24044142

**Published:** 2023-02-18

**Authors:** Yinghao Li, Qingwei Liao, Wei Hou, Lei Qin

**Affiliations:** 1Key Laboratory of Sensors, Beijing Information Science & Technology University, Beijing 100192, China; 2Key Laboratory of Technical Transformation of Pulse Electric Farm in Zhejiang Province, Hangzhou Ruidi Biotechnology Co., Ltd., Hangzhou 311100, China; 3Key Laboratory of Modern Measurement & Control Technology, Ministry of Education, Beijing Information Science & Technology University, Beijing 100192, China; 4Key Laboratory of Photoelectric Testing Technology, Beijing Information Science & Technology University, Beijing 100192, China

**Keywords:** surface plasmon, silver nanoparticles, SERS, SPR

## Abstract

A series of novel phenomena such as optical nonlinear enhancement effect, transmission enhancement, orientation effect, high sensitivity to refractive index, negative refraction and dynamic regulation of low threshold can be generated by the control of surface plasmon (SP) with metal micro-nano structure and metal/material composite structure. The application of SP in nano-photonics, super-resolution imaging, energy, sensor detection, life science, and other fields shows an important prospect. Silver nanoparticles are one of the commonly used metal materials for SP because of their high sensitivity to refractive index change, convenient synthesis, and high controllable degree of shape and size. In this review, the basic concept, fabrication, and applications of silver-based surface plasmon sensors are summarized.

## 1. Introduction

High-speed transmission of digital information is one of the urgent technical bottlenecks in the field of digital circuits. The use of photonic interconnect devices is an effective way to solve the above problem. However, due to the limitation of the optical diffraction limit, the size of conventional photonic interconnect devices is usually larger than the wavelength. The scale photonic interconnect devices are not compatible with microelectronic devices. Surface plasmon (SPs) can break the optical diffraction limit and localize the optical field in the sub-wavelength region, and the size of SP single components can be reduced to the nanometer scale, which is most likely to constitute a new generation of integrated devices—optical integrated circuits. SP is a non-radiative mode formed by the coupling of free electrons on the metal surface with incident photons, which is an electromagnetic wave propagating at the interface between the metal and the medium. SP can reduce the dimensionality of optical control from three to two dimensions to achieve effective regulation of super-diffraction limit light transmission at the nanometer scale, and at the same time, it can realize local convergence and amplification of electromagnetic energy at the nanometer scale. Due to the properties of SPP surface localization and near-field enhancement, the modulation of SPP by metal micro-nano structures, composite structures of metals and materials produces a series of novel phenomena such as optical nonlinear enhancement effects, transmission enhancement, orientation effects, high sensitivity to refractive index, negative refraction, and low threshold dynamic modulation, which make SPP useful for sensing [1,2,3,4,5], catalysis [6,7,8], waveguides [9,10], and lasers [11] and other fields to show important application prospects.

Surface plasmon, as a charge density wave, is formed by collective oscillations initiated by the mutual coupling of free electrons and photons at the metal surface, and is closely related to the properties of the metallic material itself, its structural parameters, and the dielectric environment in which it is placed [12]. Several common metallic materials (e.g., Au, Ag, Al, etc.) have surface plasmon frequencies mainly in the visible-near-infrared band, while some new materials (e.g., carbon-based materials, doped semiconductor materials, etc.) are concentrated in the near- and mid-infrared bands. The noble metals Au and Ag have relatively low free electron decay loss and can effectively avoid metal loss during the propagation or confinement of the equinoctial excitations in nano-optical applications, so they are often used as metal materials in the study of surface equinoctial properties [13]. The sensing sensitivity of silver nanoparticles for refractive index change is 1.2 times to 2 times that of gold [14,15,16,17]. In addition, the many synthesis techniques available for silver nanostructures ensure that their size and structure can be controlled, and these techniques offer theoretical possibilities for sensing [18,19]. Metal nanoparticles also enhance the Raman scattering of molecules adsorbed on their surfaces due to the equipartition excitonic properties of metals, resulting in surface-enhanced Raman spectra. In surface-enhanced Raman scattering, the enhancement multiplicity of the binding sites of silver is about two orders of magnitude higher than that of gold. However, the instability problem of silver nanoparticles largely limits their practical applications, so further compounding of silver is usually performed in applications.

Fine control of the morphology and size of nanostructures is the key to influencing the properties of surface equilibrium excitations. This paper summarizes the basic concepts of surface plasmon, their preparation processes, and their applications in biosensing.

## 2. Surface Plasmon

Noble metal nanoparticles have received much attention due to their unique optical, electronic, and mechanical properties. The most significant difference in the properties of nano-precious metals and bulk precious metals is their optical properties. The unique optical properties of metallic nanoparticles are the result of localized surface plasmon resonance excitation. Under the excitation of the incident light, the free electrons on the metal surface oscillate collectively and, together with the corresponding electromagnetic field, are able to form surface plasmons (SPs) with specific energy and frequency [20]. When the incident light has the same frequency as the surface plasmons, the two are effectively coupled, and the energy and momentum are efficiently transferred, forming a special pattern of electromagnetic field called surface plasmons resonance (SPR) as shown in Figure 1a. The SPR established in nanostructures is called localized surface plasmon resonance (LSPR), and the frequency of oscillation at which the resonance occurs is considered to be the energy of the surface plasmons. the strength of this resonance depends on the composition, shape, size, and dielectric properties of the surrounding medium of the metallic nanostructure. silver stands out among many precious metals because it has the longest electronic lifetime [21]. The size and shape of the nanoparticles are the most important factors in determining the intensity of the resonance. As shown in Figure 1b, the extinction spectra of silver nanoparticles prepared with different initial silver deposition thicknesses, the SPR appears red-shifted as the size of silver nanoparticles changes, and the extinction spectral broadening of silver nanoparticles coincides with the trend of silver nanoparticle diameter distribution, so the LSPR resonance wavelength can be controlled by controlling the silver nanoparticle size [22]. As in Figure 1c, the absorption band of silver nanoparticles is located at about 400 nm, and the colloidal suspension of silver nanoparticles is yellow, so the colorimetric detection of biomolecules can be performed by inducing the change of the position of the LSPR band [23]. As shown in Figure 1d, the absorption spectra of silver nanoparticles with different shapes differed, and the corresponding solution colors were also different [24].

Collective electron oscillations in the nanoscale region generate large field enhancements within and near nanoparticles [25], as in Figure 1e. Field enhancements are widely used in many technologies such as surface-enhanced Raman scattering (SERS) [26], metamaterials [27], photothermal therapy [28], and tip-enhanced Raman spectroscopy (TERS) [29]. The distribution of the electromagnetic field depends strongly on the size and shape of the nanoparticles used. For plasmonic nanoparticles, the strongest electromagnetic fields are located at sharp tips and edges; for aggregates, the strongest fields are located in the gaps between two or more nanostructures. Such locations are known as “hot spots”. For example [30], for silver nanostars, the strongest electromagnetic fields are located at the tips. This large enhancement of the electric field around the surface of a nanoparticle can be used for many surface-specific spectroscopic measurements, such as surface-enhanced Raman scattering, surface-enhanced hyper Raman scattering, surface-enhanced infrared absorption, or surface-enhanced fluorescence.

**Figure 1 ijms-24-04142-f001:**
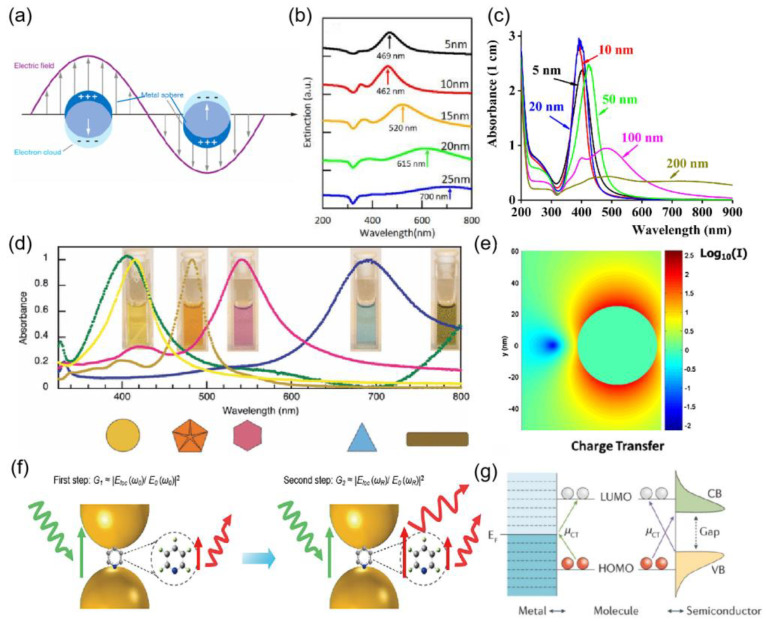
(**a**) interaction of a light wave with a spherical metallic NP causing oscillation of the electron cloud on the surface of the NP. (**b**) Extinction spectra of the Ag NPs fabricated from the different initial Ag deposition thickness. The main peak position of the extinction was indicated by the arrow. Reprinted with permission from Ref. [22]. Copyright 2021, Vacuum. (**c**) Absorbance spectra of the Ag NP of 5 nm (solid black line), 10 nm (solid red line), 20 nm (solid blue line), 50 nm (solid green line), 100 nm (solid magenta line), and 200 nm (solid olive line). Reprinted with permission from Ref. [23]. Copyright 2016, Optical Materials. (**d**) Absorption spectra for the corresponding morphologies that can be generated with the LED irradiation approach from a single precursor solution of 3 nm silver seeds. Reprinted with permission from Ref. [24]. Copyright 2012, Photochemistry and photobiology. (**e**) near-field intensity distribution around a single silver nanoparticle. Reprinted with permission from Ref. [25]. Copyright 2016, Advances in Physics. (**f**) Electromagnetic (EM) enhancement mechanism in SERS, including the two-step enhancements, illustrated in the nanogap of two metal nanoparticles. Reprinted with permission from Ref. [31]. Copyright 2018, Chemical Reviews. (**g**) Chemical enhancement through charge transfer (CT) between metal/semiconductor and the adsorbed molecule. The CT transitions (μCT) arrows show the CT directions. Red and white circles represent molecular orbitals. CB, conduction band; EF, Fermi level; HOMO, highest occupied molecular orbital; LUMO, lowest unoccupied molecular orbital; VB, valence band. Reprinted with permission from Ref. [31]. Copyright 2018, Chemical Reviews.

Surface-enhanced Raman scattering (SERS) [32] is a technique that enhances the Raman scattering of surface plasmon. Raman spectroscopy has gained interest in many fields since its discovery, such as medical diagnostics, food safety, and environmental monitoring [33,34,35]. Appropriate laser excitation produces a localized strong electromagnetic field in metal nanoparticles, which greatly enhances the Raman (and fluorescence) signal of molecules adsorbed on the surface. The magnitude of the signal enhancement is called the enhancement factor (EF) [36], denoted as:(1)$$\mathrm{EF}=\frac{I_{\mathrm{SERS}}/N_{\mathrm{Surf}}}{I_{\mathrm{RS}}/N_{\mathrm{Vol}}}$$
where N_Vol_ = _CRS_V is the average number of molecules in the scattering volume (V) for the Raman (non-SERS) measurement, and N_Surf_ is the average number of adsorbed molecules in the scattering volume for the SERS experiments. This is normally taken as representative of a substrate. More insights into this phenomenon are being sought by physicists and chemists worldwide. However, most of them agree on two common mechanisms, as shown in Figure 1, namely (f) electromagnetic enhancement (equipartition excitations) and (g) chemical enhancement (charge transfer), with the former contributing more [31].

## 3. Synthesis of Silver Nanostructure

Silver nanoparticles of different sizes and shapes are used in many fields. Various synthesis methods have undergone many improvements over time and technological developments in order to provide precise control over the shape and size of the synthesized silver nanoparticles. Silver nanoparticle fabrication is usually divided into two categories: top-down and bottom-up.

The top-down method requires the decomposition of larger structures into nanoparticles by physical means [37]. The advantage of such a method is that a significant amount of silver nanoparticles can be obtained in a short period of time. The most common preparation method in top-down is laser ablation. In [38], Using laser ablation to synthesize silver nanoparticles on cotton fabric, the resulting particles have good antibacterial properties and therefore have the potential for biosensing. Another top-down method is the vacuum sputtering method. Rezaee [39] used a DC magnetron sputtering system with rotary and turbo pumps to deposit silver nanoparticles on an alumina template. Tuchinann et al. prepared functionalized silver nanoparticles by kHz laser ablation of silver in aqueous solutions of tetramethylammonium hydroxide (TMAH) and sodium dodecyl sulfate (SDS) using nanosecond pulses [40]. The nanoparticles synthesized in this way are important analytical tools for biomedical applications. However, the top-down approach is restricted because of the use of special equipment, the high cost of synthesis, and the defects on the surface of the prepared precious metal nanoparticles.

The bottom-up approach requires the assembly of atoms or small clusters into larger nanostructures, either gradually or by the “one-pot” method. This approach facilitates the control of thermodynamic and kinetic factors in the reaction, thus effectively controlling the physical and chemical properties of the silver nanoparticles. Synthesis in the liquid phase is the most widely used synthesis method to control the shape of silver nanostructures, and obtaining anisotropic structures by controlling the shape of nanostructures is a major focus of researchers. Anisotropic particles are important for the application of plasmonic sensing, especially SERS-based sensing. Metal salt precursors are reduced in the presence of stabilizers, Ag^+^ is reduced and becomes clustered, and finally synthesized into nanostructures. The reduction of metal salt precursors can be carried out in several ways. Common bottom-up synthesis methods include chemical reduction methods, green synthesis methods, seed-mediated methods, microwave-assisted synthesis, sonochemical-mediated methods, and thermal treatment methods, As in Figure 2. The main advantages and disadvantages are shown in Table 1. Below we introduce each method individually.

### 3.1. Chemical Reduction

Chemical reduction is one of the most common ways to prepare silver nanostructures, and the materials prepared include three main categories, silver salts precursor, reducing agents, and stabilizers or capping agents. Since 1982, when Lee and Meisel first proposed [62], the use of citrate to reduce AgNO_3_ has become one of the most popular methods for preparing silver nanoparticles. During the reaction, Citrate usually has the dual role of reducing and stabilizing silver nanoparticles. The citrate ion concentration is decisive for the properties of the generated silver nanoparticles. The complexation between citrate ions and silver colloids can be promoted by increasing the concentration of citrate ions to reduce the growth of silver particles and thus obtain larger clusters. The role played by sodium citrate on the growth of silver nanoparticles was investigated experimentally and it was found that the absorption properties of Ag colloids were independent of the citrate concentration and the maximum absorbance was maintained at 400 nm. However, the reaction time for the reduction of silver ions by citrate at boiling point temperature is important to achieve complete reduction. Fewer seeds formed in the citrate reduction method and slow cluster growth contribute to the formation of larger silver nanocrystals of varying shape and size [63]. Katarzyna et al. [64] studied the effect of the combined use of tannic acid and sodium citrate on the preparation of silver nanoparticles. The synthesis using either of these acids alone does not lead to particles of uniform size and shape. The reduction of silver salts using a mixture of sodium citrate and tannic acid at 100 °C resulted in uniform particles of around 30 nm in diameter. The combined use of sodium citrate and tannic acid allows for controlled nucleation, growth, and stabilization processes, resulting in reproducible monodisperse AgNPs. Another factor affecting the synthesis process is the pH value of the reaction environment. Dong et al. [65] investigated the effect of pH on the prepared particles when using citrate as a stabilizer and reducing agent for the synthesis of silver nanoparticles. At higher pH, the shapes produced were a mixture of spherical and rod-shaped particles, while at lower pH, the shapes produced were mainly triangular and polygonal particles. Therefore, a stepwise reduction method is proposed, in which nucleation takes place in a high pH environment and growth takes place in a low pH environment, as shown in Figure 3a. The antibacterial activity of silver nanoparticles is related to their size and other properties (oxidation and release properties) [66]. In [67], Four parameters (AgNO_3_ concentration, sodium citrate (TSC) concentration, NaBH_4_ concentration, and the pH of the reaction) were improved to enhance the antibacterial activity of AgNP in the experiment, as shown in Figure 3b. It can be seen that the selection and proportion of reducing agent, heating time, and pH value play a crucial role in the morphology and size of the resulting particles when using citrate reduction to generate silver particles. In turn, the morphology and size of the particles play a decisive role in the antibacterial activity of the particles. The antimicrobial properties of silver make it a competitive candidate in the field of antimicrobial sensing [68,69]. Citrate-capped AgNPs have been the choice for SERS-based studies for a long time.

Polyols are also widely used in chemical reduction methods. Among them, ethylene glycol is the most chosen one, which acts as a solvent and reducing agent in this reaction process. Under high-temperature conditions, a precursor silver salt is added to the polyol and a stabilizer or capping agent is added to reduce Ag^+^ to the desired nanostructure. Wiley et al. [70] introduced the use of ethylene glycol as a reducing agent for silver precursors. When silver nitrate is reduced by ethylene glycol, the initially generated particles may be multiplied twinned, singly twinned, or single-crystal seeds, as shown in Figure 3c. The structure of silver nuclei varies with their size and available thermal energy, and different seeds grow into different nanostructures, therefore, the crystallinity of the seeds needs to be adjusted in the reaction as a way to control the production of specific shapes. Gautam et al. [71] used a polymer of polyvinyl alcohol (PVA) as a reducing agent to induce the reduction of Ag^+^. PVA can be used as a surface stabilizer to protect the silver particles during the reaction while preventing the appearance of agglomerates and unwanted growth. Liao et al. [72] synthesized silver nanowires with controllable morphology by an improved dual-alcohol process. Moreover, the effects of reaction temperature, different control agents, and different AgNO_3_ solution droplet acceleration on the reaction were further studied. The synthesis can be automated based on the polyol process. Wolf et al. [73] proposed an automated synthesis method to stably synthesize AgNPs with an average radius of 3 nm and 5 nm.

### 3.2. Green Synthesis

Compared to physical and chemical methods, green synthesis methods are environmentally friendly, cost-effective, and easy to synthesize NPs on a large scale, and green synthesis does not require high temperatures, high energy, and harmful chemicals [74]. Using microorganisms as reducing agents avoids the use of harmful chemicals as sealers or stabilizers. Slawson et al. [75] reported that silver nanoparticles are biocompatible in some silver-resistant bacteria. As a result, bacteria can aggregate silver on their cell walls. Pooley [76] suggested that silver could be recovered from ores using bacteria. Klaus et al. [77] proposed the synthesis of silver nanoparticles using Pseudomonas syringae AG259. The selection of suitable bacteria can effectively accelerate the production of silver nanoparticles. Shahverdi et al. [78] reported the process of using the culture supernatants of Klebsiella pneumoniae, Escherichia coli, and Enterobacter cloacae to reduce silver ions in aqueous silver nitrate solutions. The synthesis process was rapid, with nanoparticles formed within five minutes of contact between silver ions and bacteria. John et al. [79] reported the synthesis of silver nanoparticles using three bacterial strains: Rhodococcus ef1, Brevundimonas ef1, and Bacillus ef1. as reducing and sealing agents, and the synthesized silver nanoparticles are shown in Figure 4a. The sizes of the synthesized silver nanoparticles were in the range of 20–50 nm, and all the generated silver nanoparticles exhibited significant antibacterial activity.

Fungi also have the potential to synthesize silver nanoparticles because they can secrete enzymes and proteins for metal salt reduction and because fungi are generally more readily available than bacteria, and the use of fungi to synthesize silver nanoparticles also makes downstream processing for product recovery easier [80]. The size and shape of the generated material can also be adjusted by controlling the culture parameters. Ma et al. [81] used the cell-free filtrate of the fungal strain Penicillium aculeatum Su1 as a reducing agent and synthesized spherical silver nanoparticles with sizes ranging from 4–55 nm as shown in Figure 4b. The antimicrobial activity of the biosynthesized AgNPs against Gram-negative bacteria, Gram-positive bacteria, and fungi were studied by the standard Kirby-Bauer disc diffusion method. As shown in Figure 4c, the antibacterial activity of biosynthesized AgNPs against Gram-negative bacteria, Gram-positive bacteria, and fungi was higher than that of silver ions. Silver nanoparticles were synthesized using Aspergillus niger F2 by Awad et al. [82]. As shown in Figure 4d, the as-formed NPs shape was spherical and well-dispersed with sizes in the range of 2–13 nm and an average size of 8.72 ± 2.21 nm.

However, the use of microbial assistance requires additional processing steps, and microbial isolation requires additional culture maintenance. Therefore, researchers have proposed the idea of using plants and plant extracts to improve the synthesis pathway, and plant-mediated synthesis or plant synthesis involving extracts of different parts of plants (e.g., leaves, seeds, fruits, stems, roots, etc.) as reducing agents has been studied and considered as a promising strategy for nanoparticle synthesis [83]. Santos et al. [84] synthesized silver using three different citrus peel extracts, which were mixed with a solution of AgNO_3_ at a concentration of 10^−3^ mol L^−1^ and stirred at 30 °C for 60 min. The colloids prepared contain a high concentration of silver nanoparticles with a preferential spherical symmetry. The biosynthesis of silver nanoparticles using cell-free extracts of Spirulina has resulted in well-dispersed and highly stable spherical silver nanoparticles with an average particle size of 30–50 nm [85]. The use of biomolecules as templates for the synthesis of nanomaterials provides an excellent strategy for controlling and modulating their properties. Pu et al. [86] used nucleotide-based assemblies as templates for the green synthesis of silver nanoparticles, as shown in Figure 4e, and silver nanoparticles with different LSPR absorption capacities could be synthesized by changing the nucleotide assemblies.

In the past decade, green routes for AgNP synthesis have become increasingly popular due to the advantages of providing a one-step synthesis of non-toxic, eco-friendly NPs without the need for preservation or additional maintenance of cultures. However, there are some issues that need to be addressed in the green synthesis method. When using plant extracts for synthesis, the results of synthesis vary due to differences in various phytochemicals. Karthik et al. [87] using Camellia japonica leaf extract for the preparation of silver suggested that pH is an important factor affecting the morphology and size of the resulting particles and that there are differences in the pH values of different plant extracts and even of extracts from different parts of the same plant. AgNP for commercial use requires very strict uniformity of size. Yet size and shape cannot be well controlled by using green methods [88].

**Figure 4 ijms-24-04142-f004:**
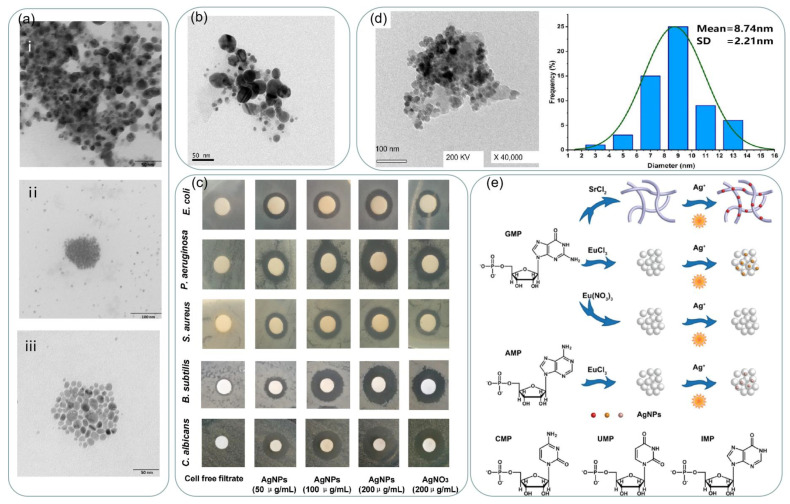
(**a**) TEM images of biosynthesized AgNPs from Rhodococcus (i), Bacillus (ii), and Brevundimonas (iii). Reprinted with permission from Ref. [79]. Copyright 2022, Marine Drugs. (**b**) TEM micrograph: scale bar 50 nm. Reprinted with permission from Ref. [81]. Copyright 2017, Materials Science and Engineering: C. (**c**) The antimicrobial effects of biosynthesized AgNPs evaluated by the standard Kirby-Bauer disc diffusion method on MHA plates. Reprinted with permission from Ref. [81]. Copyright 2017, Materials Science and Engineering: C. (**d**) Characterization of Ag-NPs formed by biomass filtrate of A. niger strain F2. TEM image and size distributions based on the TEM image. Reprinted with permission from Ref. [82]. Copyright 2022, In Journal of Fungi. (**e**) Schematic illustration of the formation of silver nanoparticles mediated by nucleotide-based assemblies. Reprinted with permission from Ref. [86]. Copyright 2018, ACS Applied Materials and Interfaces.

### 3.3. Seed-Mediated Growth

Seed-mediated growth processes are widely used because of the high degree of control over the size, shape, and structure of the generated particles. Seed-mediated methods can be further divided into two types of growth: homogeneous epitaxy and heterogeneous epitaxy [89].

Homogeneous epitaxial growth refers to the growth of crystalline seed crystals containing the same metal as that deposited on the seed. Lin et al. [90] prepared silver nanospheres and silver nanotubes using a homogeneous epitaxial seed-mediated method. The synthesis process is illustrated in Figure 5a. Hegde et al. [91] proposed a method for the synthesis of triangular nanoplates using the seed-mediated method and cetyltrimethylammonium bromide (CTAB) as a single capping agent at a very low concentration of 0.4 mM. Wang et al. [92] developed a facile synthesis method based on seed-mediated growth using glucose as a reducing agent, where secondary nucleation is prevented during seed-mediated growth due to its weak reducing ability, and the size of silver nanoparticles can be continuously adjusted by continuously adding reactants. The rate of reductant addition also affects the growth, if the rate of reductant addition is too fast so that the consumed reagent is less than the added reagent, the concentration per unit volume will increase or even exceed the critical concentration for secondary nucleation and new nucleation will occur. Using this method, silver nanoparticles with diameters ranging from about 20 nm to 120 nm can be obtained.

Unlike homogeneous epitaxial growth, heterogeneous epitaxial growth requires the deposition of metals different from the seeds, and usually, a gap in the lattice constants of the seeds and the deposited metals are required. Tian et al. [93] proposed a method to obtain Au@Ag nanocubes by epitaxial growth using gold nano octahedra as cores, and the obtained nanocubes are shown in Figure 5b. Tharion et al. [94] reported a method to obtain highly monodispersed silver nanoparticles using heterogeneous epitaxial growth of gold seed particles, where silver was deposited on the gold seed particles by Torrhen reaction. The presence of seed particles was well controlled, achieving a standard deviation of ≤11% in size. Nain et al. [95] used gold seeded nanoparticles of size ~2.4 nm as a template to prepare silver nanoparticles using heterogeneous epitaxial growth. Areesha et al. [96] demonstrated a seeded approach for the in-situ growth of Ag nanostructures directly on filter paper. Multiple growth steps were used to increase the size of Ag NPs grown from Au seeds as shown in Figure 5c. After the synthesis of Ag NPs through one growth cycle, very small NPs are distributed throughout the substrate as shown in Figure 5d. However, the particle size increases with each successive cycle, and after four growth cycles, relatively much bigger particles can be seen on the surface, as shown in Figure 5e.

Seed-mediated growth offers a simple and robust method for engineering the surface of nanocrystals. The facet can be altered by using a suitable capping agent. For example, Zeng et al. [97] found that nano-octahedra surrounded by {111} facets and nanocubes/nanorods covered by {100} facets could be selectively synthesized using citrate and PVP as capping agents. The size of nanocrystals can be conveniently tuned and tightly controlled in seed-mediated growth by varying the ratio of precursor to seed under suitable thermodynamic or kinetic conditions. For example, Jeon et al. [98] produced 2 μm cubes with precise cubic geometry by controlling the reaction atmosphere and the use of hydrochloric acid.

**Figure 5 ijms-24-04142-f005:**
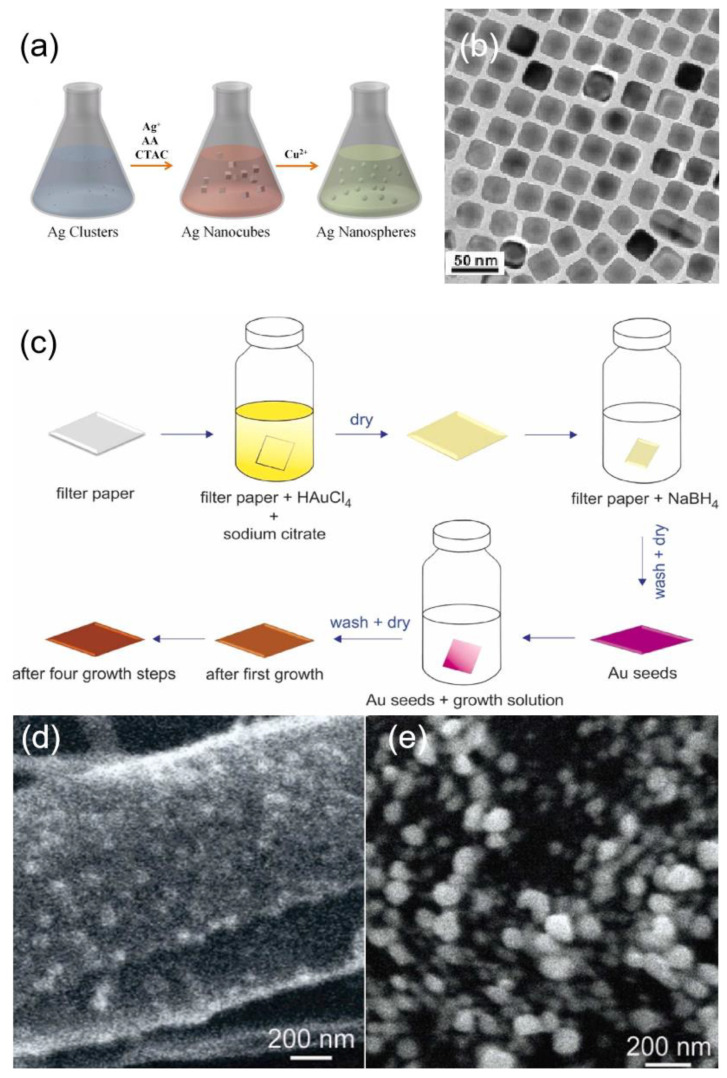
(**a**) Schematic illustration of the synthesis process of Ag NSs using a seed-mediated growth coupled with a chemical etching method. Reprinted with permission from Ref. [90]. Copyright 2018, Langmuir. (**b**) TEM image of Au@Ag nanocubes. Reprinted with permission from Ref. [93]. Copyright 2008, Journal of the American Chemical Society. (**c**) Schematic representation of the growth of Ag NPs on Au seeds. Reprinted with permission from Ref. [96]. Copyright 2022, Vibrational Spectroscopy. (**d**) substrate with 1st growth cycle. Reprinted with permission from Ref. [96]. Copyright 2022, Vibrational Spectroscopy. (**e**) SEM image of the substrate after the 4th growth cycle. Reprinted with permission from Ref. [96]. Copyright 2022, Vibrational Spectroscopy.

### 3.4. Microwave-Assisted Growth

Microwave-assisted growth is receiving increasing attention due to its efficient and environmentally friendly nature. Compared to other conventional methods, microwave radiation requires less energy and can produce nucleation sites in solution at a very fast rate, thus significantly increasing the reaction rate [99].

Anjana et al. [100] reported a rapid and green microwave-assisted synthesis method using Cinnamomum cinereum (C. cinereum) as a reducing and capping agent to prepare silver NPs in a crystalline and spherical shape with an average size of 19.25 nm. Strapasson et al. [101] used glycerol as a solvent and reducing agent and food-grade corn starch as a stabilizer to prepare silver nanoparticles by microwave radiation and the reaction principle is shown in Figure 6a. The synthesis method was reproducible and the silver NPs synthesized after 320 days were highly stable as shown in Figure 6b. Revnic et al. [102] reported a very simple, rapid, and reproducible microwave-assisted preparation of anisotropic silver nanostars (AgNS), which consist of a central nanoparticle interconnected with several highly one-dimensional single arms having a star shape as shown in Figure 6c,d. Microwave-assisted synthesis of nanostars was performed in sealed-bottom flasks at very short time intervals (less than 3 min), and the only chemical precursors in the wet chemistry procedure for synthesis were silver nitrate and trisodium citrate, and the plasma substrates synthesized by this method were able to produce strong SERS spectra.

One of the advantages of microwave-assisted synthesis is its ability to increase the reaction rate. Li et al. [103] used solvent-free microwave extraction (SFME) to shorten the reaction time from 3 h to 15 min as shown in Figure 6e. Another advantage is that the shorter reaction time results in less by-product formation and therefore higher yields [104]. For example, microwave-assisted effective improvement of the enzymatic digestion of cellulose increases the yield of glucose, as shown in Figure 6f [105].

**Figure 6 ijms-24-04142-f006:**
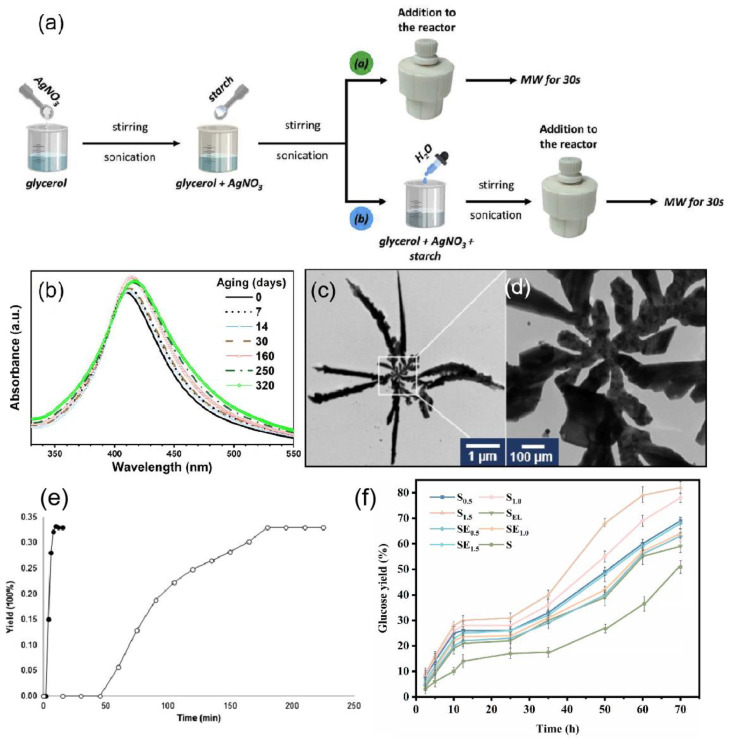
(**a**) Schematic representation for the syntheses of Ag NPs, where the solvent is a. pure glycerol and b. glycerol/water mixtures. MW: microwave. Reprinted with permission from Ref. [101]. Copyright 2021, International Journal of Hydrogen Energy. (**b**) Aging of Ag NPs. Reprinted with permission from Ref. [101]. Copyright 2021, International Journal of Hydrogen Energy. (**c**) TEM image of a typical individual silver nanostar. (**d**) TEM image of the central zone of the nanostar. Reprinted with permission from Ref. [102]. Copyright 2022, International Journal of Molecular Sciences. (**e**) Yields as a function of time for extractions of essential oil from rosemary leaves by microwave hydro-diffusion and gravity (MHG) (●) and hydro-distillation (○). Reprinted with permission from Ref. [103]. Copyright 2013, TrAC Trends in Analytical Chemistry. (**f**) Glucose yields of enzymatic hydrolysis of the cellulose-rich residues obtained from the integrated process. Reprinted with permission from Ref. [105]. Copyright 2022, Energy.

### 3.5. Sonochemical-Mediated Synthesis

Another synthetic modality that has received attention is the sonochemical mediated synthesis, where the commonly used excitation source for sonochemistry is mainly high-energy ultrasound, usually in the frequency range of 16 KHz–5 MHz, and most reactions are carried out in the liquid phase [106]. Ultrasound itself cannot act directly on molecules but affects them through the physical effects of the surrounding environment in turn. When silver nanoparticle is prepared by the sonochemical method, the compression process of cavitation bubbles is rapid, and the heat in the cavitation bubbles is too late to be transferred to the liquid medium, so high temperatures are generated instantly in the hot spot region, and the liquid walls around the collapsed bubbles compress the substances contained in the bubbles, creating a good reflective environment for the chemical reaction.

Zhou et al. [107] reported a simple preparation method involving ultrasonic irradiation and glutathione as a stabilizer for the fabrication of blue luminescent silver nanoclusters. Kumari et al. [108] reported a simple method for the synthesis of silver/graphene nanocomposites by sonochemical method using sodium citrate as a reducing agent, which reduces silver ions to silver nanoparticles and forms spherical nanoparticles with an average particle size of 20 nm on graphene sheets. SEM images of different GO layers and GO surface precipitation are shown in Figure 7a,b, respectively, and the morphology of the GO-Ag composite structure and the lamellar structure at the embedding of silver nanoparticles examined using TEM are shown in Figure 7c,d. Particles with good dispersion can be obtained by precise control of the reaction conditions. Darroudi et al. [109] investigated the effect of reducing agent concentration, Ag^+^ concentration, sonication event, and sonication amplitude on the size of the resulting particles. Spherical silver nanoparticles with good dispersion were finally obtained, and the average particle size of the particles was about 3.5 nm, as shown in Figure 7e. Murgunde et al. [110] introduced a rapid method for the synthesis of silver foams. The synthesis method was performed by the ultrasonic pulsing of the reactants to reduce Ag^+^ to Ag on the silicon surface. The study by Jameel et al. [111] focuses on a facile and environmentally friendly process to synthesize uniformly cubic-shaped silver nanoparticles (AgNPs) with an ultrasonic assist in a short time. Figure 7f illustrates the proposed and possible mechanism of the sonochemical process that reduces Ag^+^ to Ag^0^. The resulting non-spherical silver nanoparticles are shown in Figure 7g.

Sonochemical synthesis methods have the unique advantage of controlling the size of the generated particles by controlling the ultrasonic frequency [112]. In general, sonochemical synthesis produces spherical metal nanoparticles, and thus the sonochemical process had been limited in preparing other metal nanostructures (e.g., nanorods, nanowires, etc.). To obtain other structures, it is usually necessary to add other substances as structure pointers [113].

## 4. Applications of Silver-Based Plasmonic Sensing

Plasmon sensor, as a kind of optical sensor, has high sensitivity and recovery ability and does not require many expensive professional instruments. Here we will mainly focus on SPR sensing and SERS sensing as shown in Figure 8.

### 4.1. Silver-Based Surface Plasmon Resonance Sensing Applications

SPR and LSPR sensors are based on the modulation of the refractive index of the sensing layer around the metal nanostructure due to the physicochemical interaction with the analyte [114]. This section describes the application of silver-based surface plasmon resonance sensors in three fields: food detection, environmental monitoring, and biosensing.

#### 4.1.1. Food Detection

An important area of SPR-based sensor applications is food detection. SPR aids in label-free detection of various components such as adulterants, antibiotics, biomolecules, genetically modified foods, pesticides, insecticides, herbicides, microorganisms, and microbial toxins in food and assures safety [115]. Testing the purity of food is an important part of ensuring the nutritional value of food, and accurate and rapid testing methods are necessary. Vikas et al. [116] designed and developed an SPR sensor using silver and silver graphene oxide (GO) for the detection of glucose, fructose, and other adulterants in honey. Bacteria such as Pseudomonas can be spread through food [117], so testing for bacteria in food can be effective in ensuring food safety. Mudgal et al. [118] proposed a highly sensitive SPR biosensor based on silver, barium titanate (BaTiO_3_), graphene, and an affinity layer for the detection of Pseudomonas aeruginosa. The proposed structure was compared with a contemporary surface plasmon resonance (SPR) biosensor for the detection of Pseudomonas aeruginosa, which showed better performance. Food testing is characterized by high quantities and high repeatability, so the cost of money and time needs to be taken into account. Paper-based sensors open up new ways to produce disposable, low-cost devices. In [119], a platform for selective recognition of chiral analytes was obtained by embedding silver nanoparticles synthesized in situ into transparent nanopapersin. The sample volume required to use this identification platform is small (50 μL) as shown in Figure 9a. Figure 9b shows the UV-Vis spectra of the micrographeme containing AgNP embedded in nanopaper. The enantiomeric percentage of D-cysteine was operated in the range of 5% to 100% range, with the absorption peaks red-shifted (to 600 nm) at different enantiomeric percentages of D-cysteine, indicating that the developed microcrystalline membrane platform is a simple and rapid method for sensitive and selective optical sensing of D-/L-cysteine enantiomeric percentages. From this, the type of bacteria can be determined [120]. It can be seen that portable, cost-effective, and rapid sensing methods are required for food detection. There are already some paper-based sensors being proposed by researchers to control the cost. However, sustainable sensors have rarely been proposed. Further research on sensors with sustainable applications is a challenge for future food detection work.

#### 4.1.2. Environmental Monitoring

Due to industrialization, clean water resources are becoming increasingly scarce. Water contaminants can be organic matter, microorganisms, metal particles, and others, which are extremely harmful to humans [121]. The detection of contaminants using silver based SPR sensors has become a popular application currently. In a study [122], an optical fiber plasmonic element sensor based on a silver thin film was proposed for the detection of the phenolic analyte catechol, which is an organic substance that seriously pollutes the environment. The detection principle is shown in Figure 10a. The sensor relies on the interaction of catechol with cetyltrimethy lammonium bromide (CTAB) functionalized zinc oxide/carbon nanotubes (ZnO/CNTs) nanocomposites coated on a silver film. The probe exhibited reproducibility and high selectivity in the concentration range of catechol from 0 to 100 µM. Rajkumar et al. [123] used the avocado extract to synthesize silver nanoparticles which could be used to detect potentially hazardous Hg^2+^ ions. The schematic diagram of the detection mechanism of Hg^2+^ by AgNPs is shown in Figure 10b. The sensing mechanism is owing to the redox chemical reaction between Ag^0^ and Hg^2+^. The SPR band vanished which confirmed the high sensitivity towards Hg^2+^ ions. For all other investigated metal ion solutions, there was no change in the color of AV-AgNPs. In [124], a new, highly specific, and simple colorimetric assay for the chemosensing of sulfide ions (S^2−^) in environmental water samples has been developed, by using cysteine-functionalized colloidal silver nanoparticles (Cys-AgNPs). The sensor has highly specific and high sensitivity as shown in Figure 10c. Air pollution is also an important issue facing us today, so the detection of pollutants in the air is also in need of improvement. In addition to water pollution, air pollution is also an important issue in society. Recently, Pan et al. [125] designed a portable detection device that uses a colorimetric method for the selective monitoring of lead (Pb) in the air. They embedded silver nanoparticles prepared from citrate into cellulose paper strips and monitored Pb by silver nanoparticles showing a distinct localized surface plasmon resonance (LSPR) peak in the blue region, with increasing Pb concentration, the LSPR band decreases, and a new peak appears in the green region of the spectrum as larger aggregates form. The device monitors these spectral changes and gives the amount of lead present in the sample accordingly. The most important quality for sensors used in environmental monitoring is to have a strong specificity to the substance being detected because the water or air components to be detected are more complex. Future environmental monitoring sensors will need to control costs and increase portability as much as possible while ensuring their specificity.

#### 4.1.3. Biomedical Sensing

LSPR biosensors also have great potential for applications in the biomedical field [126]. Chen et al. [127] developed a reflective localized surface plasmon resonance (LSPR) fiber optic sensor based on silver nanoparticles (Ag-NP). Figure 11a shows a schematic diagram of the sensing method of this sensor. The sensor probe was further modified by the antigen. The fabricated biosensor probe could be used to monitor the interaction of an antigen and antibody. As shown in Figure 11b, the immobilization of anti-human IgG on the sensor-based surface was observed in 35 min. Austin et al. [128] reported the development of a biosensor that utilizes the localized surface plasmon resonance (LSPR) effect of silver nanostructures to detect dengue NS1 antigen by thermal annealing of metal films, and the detection principle is shown in Figure 11c. Yang et al. [129] have developed a facile DNA detection platform based on a plasmonic triangular silver nanoprism etching process, in which the shape and size of the nanoprisms were altered accompanied by a substantial surface plasmon resonance shift. Figure 11d schematically depicts the working principle of the assay. Figure 11e shows the UV-Vis absorption spectra of the silver nanoprism in the presence of different DNA. the largest SPR peak shift was observed when the target DNA was perfectly matched. however, when the mismatched sequences were introduced in the target DNA, fewer peak shifts occurred. The results demonstrate the excellent selectivity of the assay. The localized surface plasmon resonance of the nanostructure has a symbiotic relationship with the 3D photonic stop band, leading to highly tunable characteristics [130,131]. The interaction of silver nanomaterials and quantum dots has been used in sensing [132]. Silver-based sensors perform well in the biomedical field. The good biocompatibility of silver makes it well-suited for biosensors [133]. However, the toxicity of silver is an obstacle to its potential application in biosensing. Further research on the mixing of silver with other materials is needed to overcome this obstacle.

### 4.2. Silver-Based Surface Enhanced Raman Scattering Sensing Applications

SERS is a technique to enhance the Raman scattering of molecules supported by some nanostructured materials. After the discovery of SERS, Raman spectroscopy has gained attention in many fields. In SERS detection applications, silver nanomaterials usually need to be compounded with other materials, and there are two ways of compounding, the first one is to use other materials as the surface to grow or adsorb silver nanostructures [134]; the other one is to wrap the silver nanostructures with other materials, thus avoiding their contact with the outside world [135].

#### 4.2.1. Supported Structure

Previous studies have shown that the shape, morphology, space, assembly, and distance of the substrates may affect the SERS activity [136]. Some researchers have found that silver and other metals can be used in conjunction with each other to obtain better Raman enhancement [137]. Silver can also be combined with other materials to obtain better results. Wei et al. [138] proposed a silver nanoparticle (NP) modified graphene nanoribbon (GNR) to be used as a high-performance shared substrate for surface-enhanced Raman and infrared absorption spectroscopy (SERS and SEIRAS) as shown in Figure 12a, and the preparation process is shown in Figure 13a. The SERS enhancement factor of the order of 10^5^ can be obtained by optimizing the size of silver nanoparticles in this experiment. Therefore, when silver-based SERS sensing is used in food safety, environmental testing, biomedical and other applications, more attention should be paid to the material and characteristics of the substrate.

Silver-based SERS sensing is widely used in the field of food detection. The study by Sun et al. [144] presented a simple, effective, and reproducible SERS substrate for the sensitive detection of melamine and thiram in food samples. A three-dimensional gold nanodendrites (AuNDs) decorated by silver nanoparticles (AgNPs) via electrodeposition has been synthesized as a hybrid SERS-active substrate. The synthesized structure is visible in Figure 13b and the EDX spectrum proves the co-existence of Ag and Au in the structure. The SERS signal reflects that the structure exhibits good detection ability for thiram. Chen et al. [145] Developed an Ag/Nanocellulose fibers SERS substrate (Ag@NCF substrate). They detected malachite green (MG), Enrofloxacin (ENRO), and thiram, thiabendazole (TBZ) on fish and pear respectively as shown in Figure 13c. It is proved that the Ag@NCF substrate has broad adaptability to real-world matrices in different textures and enables sensitive, on-site, in-situ SERS detection with a portable Raman spectrometer. Attaching silver to flexible fabrics helps achieve flexibility in information acquisition [146]. Gao et al. [147] used magnetron sputtering to coat Ag on cotton fabric and used it as a SERS substrate as shown in Figure 13d. This substrate exhibited good signal stability. Its limit of detection for thiram is far lower than the national standard. Using in-situ growth methods results in better adhesion of silver. Better electrical conductivity and more stable electrical properties compared to conductive fabrics prepared using the dip-coating process [148].

**Figure 13 ijms-24-04142-f013:**
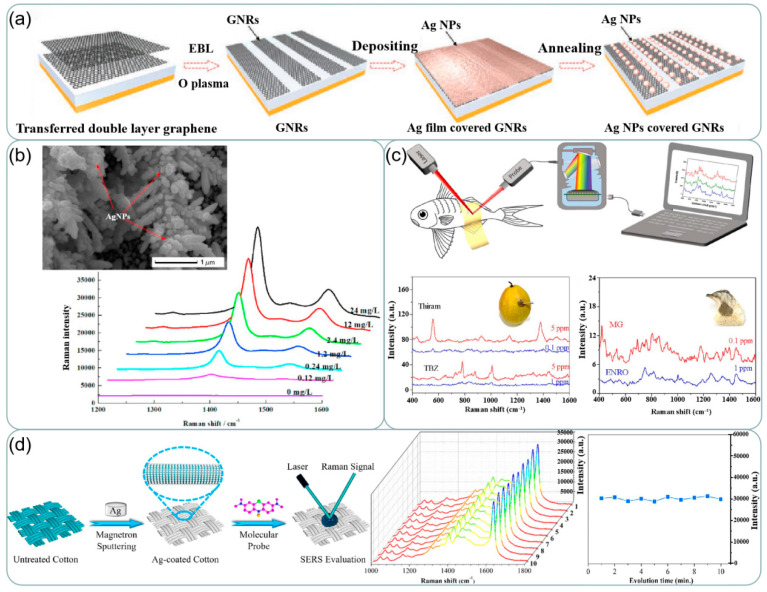
(**a**) Fabrication processes of the shared substrate. Reprinted with permission from Ref. [138]. Copyright 2021, Small. (**b**) SEM images and EDX spectroscopy of Ag-AuNDs substrate. SERS spectra of thiram with a concentration of 0–5 mg/L and 0–24 mg/L. Reprinted with permission from Ref. [144]. Copyright 2021, Food Control. (**c**) In-situ SERS detections of hazardous residues on fish by a ‘paste-test’ method. Raman spectra of thiram and TBZ detected on pears. Raman spectra of MG an ENRO detected on fishes. Reprinted with permission from Ref. [145]. Copyright 2021, Applied Surface Science. (**d**) Schematic of preparation process of superhydrophobic Ag-coated cotton fabric substrate and SERS evaluation of prepared substrates. Changes in Raman spectra of MB with time on the same area of AC100 plotted at 1 min intervals for 10 min, and intensity of peaks of Raman signals. Reprinted with permission from Ref. [147]. Copyright 2020, Applied Surface Science.

An increasing number of silver-based SERS sensors are also present in the environmental detection field. In [149], Ag_3_PO_4_ microcubes were synthesized as an efficient, ultrasensitive, and recyclable SERS substrate for the detection of heavy metals. The fabricated Ag_3_PO_4_ microcubes substrate provided a remarkable SERS sensitivity with analytical enhancement factor (AEF ~ 10^10^), uniformity (<10%), and stability towards Hg^2+^ and Pb^2+^ ions. Butmee et al. [150] reported a highly sensitive SERS-based sensor for the determination of glyphosate herbicide based on enveloping the AgNPs decorated TiO_2_ NTs arrays with rGO, denoted as TiO_2_ NTs/AgNPs-rGO as shown in Figure 14a. Figure 14b shows how the SERS intensities increase with increasing glyphosate concentrations. According to the symmetric stretching of the PO_2_ functional group at 920 cm^−1^, the SERS signal exhibited a good linear relationship with the logarithmic value of glyphosate concentration in the range from 0.005 to 50 mg/L (Figure 14c). He et al. [151] developed a novel SERS-active plasmonic silver membrane (HOX@Ag-PVDF) for the rapid, selective, and ultrasensitive detection of Cu^2+^, where hydrophobic hydroxyoxime (HOX) molecules served as the Raman reporter and target catcher (Figure 14d). HOX can serve as a Raman reporter for monitoring spectral changes after trapping Cu^2+^. As shown in the inset of Figure 14e, the color of HOX@Ag-PVDF changes from brown to dark green after capturing Cu^2+^. The capture mechanism of HOX@Ag-PVDF for Cu^2+^ is shown in Figure 13f.

SERS detection has also received attention in the field of biomedical [152]. Zhang et al. [139] fabricated a silver-modified zinc oxide array chip with the structure shown in Figure 12b, which can be used to detect opiates with high accuracy and sensitivity. Oxycodone concentrations of 500 μg/mL, 10 μg/mL, 1 μg/mL, and 100 ng/mL showed the highest characteristic peaks in the Raman spectroscopy with a common baseline at around 1360 cm^−1^ and 1590 cm^−1^ in Figure 15a. In [140], Silver nanowire ink was written on the surface of drawing paper by automatic writing method as shown in Figure 12c. The prepared arrays exhibited a strong SERS response to crystal violet and a low detection concentration of 10^−15^ mol/L was obtained. In [153], 4-mercaptophenylboronic acid (4-MPBA) was modified on the surface of silver nanoparticles (AgNPs) under mild conditions to obtain a boronic acid-functionalized SERS substrate for the detection of fructose in artificial urine. The detection mechanism was based on the deboronization reaction of 4-MPBA on the surface of AgNPs (Figure 15b), which was induced by fructose, and the Raman signal of the generated thiophenol (TP) molecules was significantly enhanced by the silver nanoparticles, with a linear increase in SERS peak intensity at 1570 cm^−1^ as shown in Figure 15b. Recently, a novel SERS/electrochemical dual-mode biosensor based on multi-functionalized molybdenum disulfide nanosheet (mF-MoS_2_ NS) probes and SERS-active Ag nanorods (AgNRs) array electrode was proposed for in situ dual-mode detection of gastric cancer-related miR-106a [154]. Figure 15c shows a schematic illustration of the sandwich-structured sensing mechanism of the SERS/electrochemical dual-mode biosensor for detecting gastric cancer-related miR-106a.

#### 4.2.2. Core-Shell Structure

In the second type of composite, the use of silver nanomaterials and other materials conforming to the composition of a core-shell structure is the most popular type of composite for current applications, and the most important purpose of this type is to improve the sensitivity of SERS detection [155,156,157]. Chen et al. [141] synthesized several layers of MoS_2_ directly on silver nanoparticles by thermal decomposition, resulting in a MoS_2_/AgNPs hybrid system for surface-enhanced Raman scattering as shown in Figure 12d. The nano-gap between the two metallic nanostructures has been shown to be effective in concentrating the incident electromagnetic field into a small space. As a result, the resulting strong field localization can greatly enhance SERS. Jin et al. [158] prepared Ag-Au-Ag structures with nanogaps, which can be tuned by ion-controlled size and distribution of sputtered Au nanoparticles, and the SERS performance is enhanced due to the presence of nanogaps with significantly enhanced electric field strength between the two Ag-NP layers. 

Another advantage of core-shell structures is that the shells are more easily modified than silver nanostructures grown directly on the substrate. Xin et al. [159] used a low-temperature heating and stirring method to prepare Ag@C core-shell nanoparticles, and then the 24 nm silver nanoparticles were attached to the Ag@C particle surface by different concentrations of polyethyleneimine (PEI), and finally, Ag@C@Ag hybrid nanoparticles were prepared. Such particles have excellent and stable optical properties and can be used as SERS substrates in the field of ultrasensitive spectroscopy. The modified core-shell structure has better physical and chemical properties. 

SERS of core-shell particles is of interest for its high sensitivity in food safety, environmental monitoring, and biomedicine. Silver shell was formed on the core gold nanoparticles protected with inositol hexaphosphate (IP6), designated as Ag@IP6@AuNPs [160]. This SERS substrate is employed to successfully detect trace Penicillin G residue in milk with a wide linear quantitative range from 10^−5^ to 10^−11^ mol/L and the limit of detection at 10^−12^ mol/L as shown in Figure 16a. In [142], a core–shell-satellite structured SERS substrate Fe_3_O_4_@MIL-100(Fe)@Ag nanoparticles (FMAs) have been developed. It has multiple functions such as adsorption, detection, degradation, and recovery. The schematic of the nanoparticle is shown in Figure 12e. The structure allows the detection of crystalline violet in farmed water and tilapia. Bao et al. [143] introduced aluminum and iodide ion-modified silver nanoparticles to meet the requirements of Raman detection, and the structure is shown in Figure 12f, which can be used to analyze proteins with different PIs, as shown in Figure 16b,c for six proteins with different amounts of Trp and Phe (lysozyme, α-casein, insulin, myoglobin, peroxidase, and BSA), and Figure 16d shows the relative Raman intensity of 752/1004 with the ratio of Trp/(Trp and Phe), and it was found that the relative intensity from 752 to 1004 cm^−1^ correlates well with the ratio of Trp to (Trp+Phe), so the silver nanoparticles modified by aluminum and iodide ions have good potential for quantitative and multiplex analysis.

## 5. Summary and Future Prospect

An increasing number of nanotechnologies are choosing to introduce plasmonic due to their ability to moderate and manipulate light. In this review, we have specifically chosen silver for discussion. Five synthesis methods of silver nanostructures are reviewed here, and the advantages and disadvantages of each are discussed. Then, the silver-based SPR sensing and SERS sensing applications in food safety, environmental monitoring, and biomedicine are reviewed, respectively. From the introduction of this paper, it is clear that the overall trend of silver nanostructure synthesis methods is toward green and reproducible directions. Silver-based SPR sensing is receiving more and more attention, but at present, it is mostly used in food detection and other fields as a one-time detection method, and its reusability and durability are still in urgent need of a solution. Since Raman spectrometers are usually large and expensive, they limit the application of SERS-based medical devices. The current direction of SERS development is mainly to develop paper-based substrates, palm-sized spectrometers, etc. The rapid development of silver plasmonic proves its usefulness for the development of society and inspires researchers around the world to gain a deeper understanding of it.

## Figures and Tables

**Figure 2 ijms-24-04142-f002:**
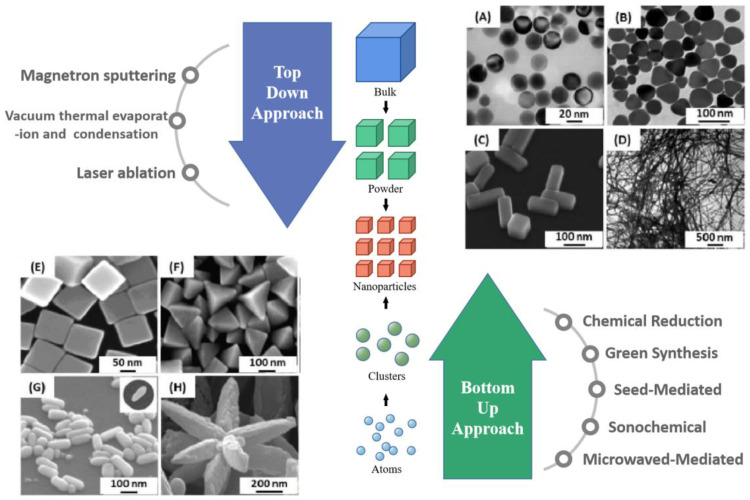
Top-down and bottom-up approach for fabricating Ag NPs. Major synthetic approaches for the fabrication of Ag NPs. TEM images of silver nanoparticles with different shapes: (**A**) nanospheres Reprinted with permission from Ref. [41]. Copyright 2004, Nano Letters. (**B**) nanoprisms Reprinted with permission from Ref. [42]. Copyright 2012, Colloids and Surfaces A. (**C**) nanobars Reprinted with permission from Ref. [43]. Copyright 2007, Nano Letters. (**D**) nanowires Reprinted with permission from Ref. [44]. Copyright 2002, Advanced Materials. SEM images of (**E**) nanocubes, (**F**) pyramids Reprinted with permission from Ref. [45]. Copyright 2006, The Journal of Physical Chemistry B. (**G**) nanorice Reprinted with permission from Ref. [43]. Copyright 2007, Nano Letters. and (**H**) nanoflowers Reprinted with permission from Ref. [46]. Copyright 2011, Materials Research Bulletin.

**Figure 3 ijms-24-04142-f003:**
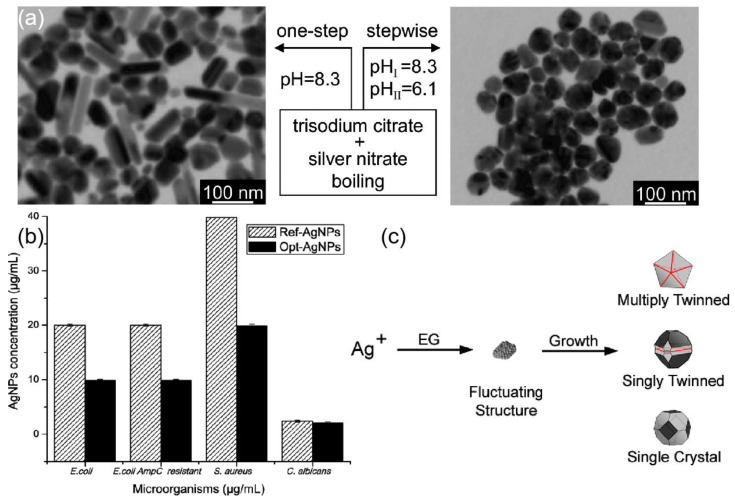
(**a**) A stepwise reduction method, in which the nucleation and growth processes were carried out at high and low pH, respectively, was proposed for the syntheses of spherical silver nanoparticles. Reprinted with permission from Ref. [65]. Copyright 2009, The Journal of Physical Chemistry C. (**b**) Optimization of silver nanoparticle synthesis by chemical reduction and evaluation of its antimicrobial and toxic activity. Reprinted with permission from Ref. [67]. Copyright 2019, Biomaterials research. Copyright 2019, Biomater Research. (**c**) Reduction of Ag^+^ ions by EG leads to the formation of nuclei. The structure of nuclei fluctuates depending on their size and the thermal energy available. Reprinted with permission from Ref. [70]. Copyright 2007, Accounts of Chemical Research.

**Figure 7 ijms-24-04142-f007:**
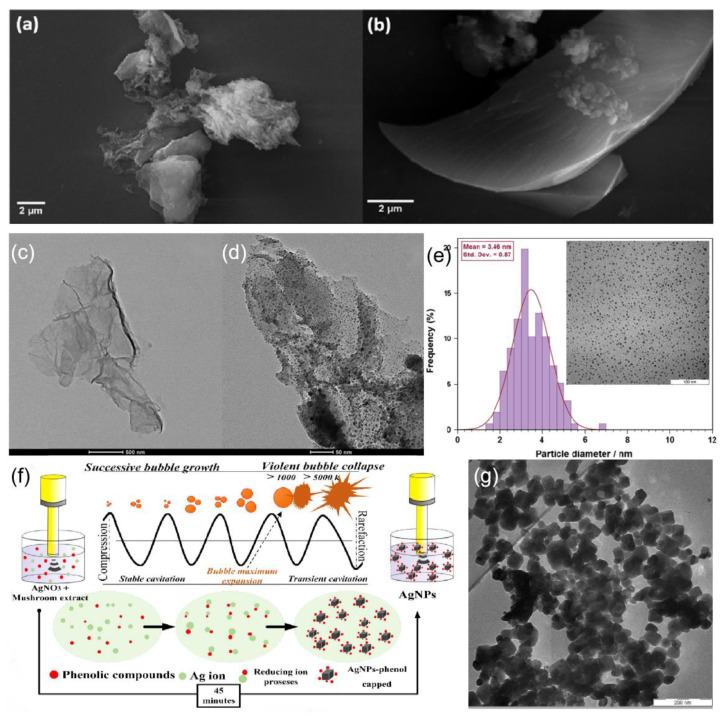
SEM micrographs of (**a**) GO and (**b**) GO-Ag nanocomposite and (**c**) EDX of GO-Ag nanocomposite. TEM micrographs of (**c**) GO and (**d**) GO-Ag nanocomposite. Reprinted with permission from Ref. [108]. Copyright 2020, ACS Omega. (**e**) TEM image of Ag-NPs synthesized under ultrasonic irradiation for 45 min (AgNO_3_ = 0.1 M, gelatin = 1%, and amplitude = 50). Reprinted with permission from Ref. [109]. Copyright 2012, Materials Letters. (**f**) Schematic illustration of the synthesis process of cubic AgNPs using sonochemical process. Reprinted with permission from Ref. [111]. Copyright 2022, Surfaces and Interfaces. (**g**) TEM image of Ag cubic. Reprinted with permission from Ref. [111]. Copyright 2022, Surfaces and Interfaces.

**Figure 8 ijms-24-04142-f008:**
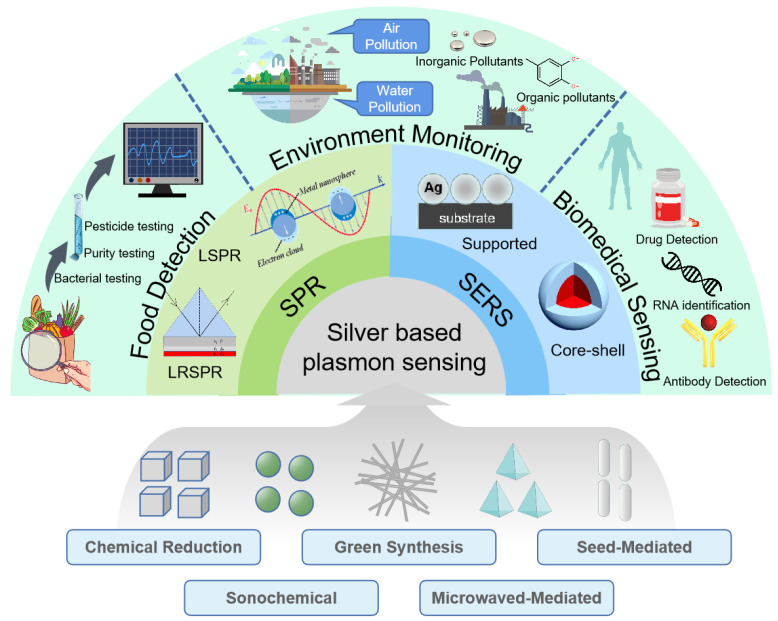
Applications of silver-based plasmon sensing in three fields: food detection, environment monitoring, biomedical sensing.

**Figure 9 ijms-24-04142-f009:**
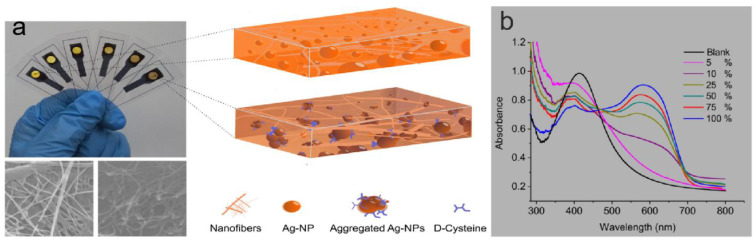
(**a**) The microcuvette modulation for spectrophotometric measurements at the solid platform for the same enantiomeric percentage range of D-cysteine. Reprinted with permission from Ref. [119]. Copyright 2018, Talanta (Oxford). (**b**) The recorded UV-Vis spectra for the corresponding microcuvettes. Reprinted with permission from Ref. [119]. Copyright 2018, Talanta (Oxford).

**Figure 10 ijms-24-04142-f010:**
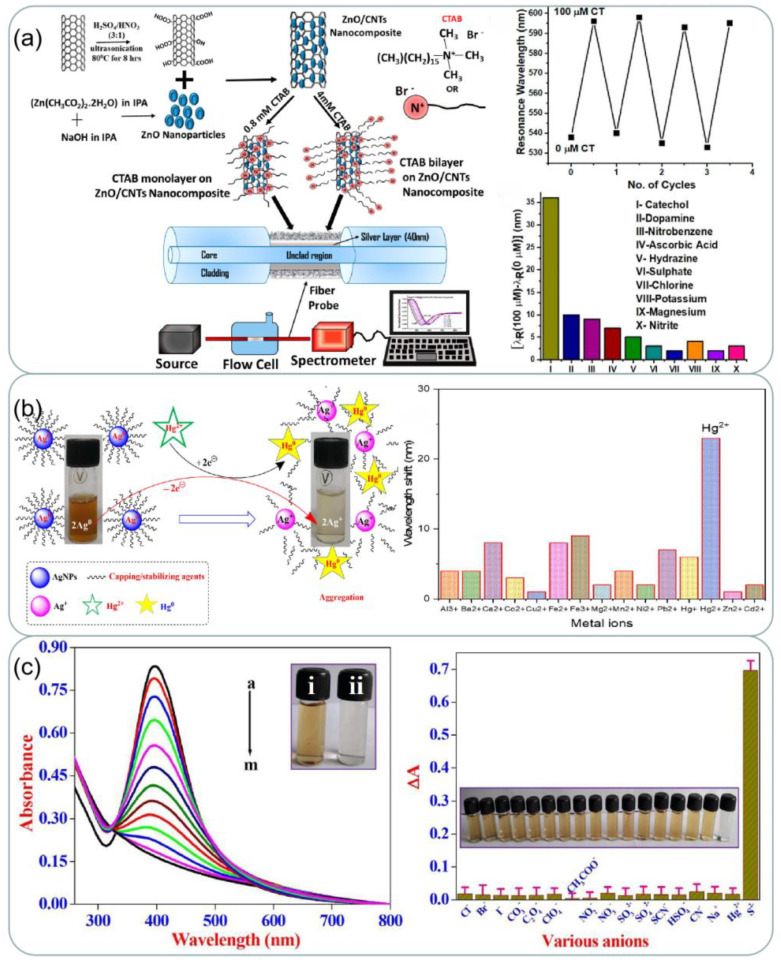
(**a**) Schematic of synthesis of functionalized CNTs and ZnO nanoparticles, attachment of ZnO nanoparticles to CNTs walls, CTAB functionalization of ZnO/CNTs nanocomposite for two concentrations, fiber probe, and experimental set-up. Results of repeatability and selectivity for the finalized. Reprinted with permission from Ref. [122]. Copyright 2020, ACS Applied Nano Materials. (**b**) Schematic illustration of sensing of Hg^2+^ by AV-AgNPs. UV-vis spectra recorded upon 1 mL addition of AV-AgNPs with different metal ion solutions (500 μM, 2 mL). Reprinted with permission from Ref. [123]. Copyright 2022, Journal of Molecular Liquids. (**c**) SPR band changes of Cys-AgNPs with different amounts of S^2−^ ion and inset shows photographic image of (i) Cys-AgNPs in the (ii) final addition S^2−^ ions. Relative absorption intensities and the colorimetric response of Cys-AgNPs towards S^2−^ ions as well 100-fold higher concentrations of different interfering inorganic anions and few cations. Reprinted with permission from Ref. [124]. Copyright 2022, Microchemical Journal. Copyright 2022, Microchemical Journal.

**Figure 11 ijms-24-04142-f011:**
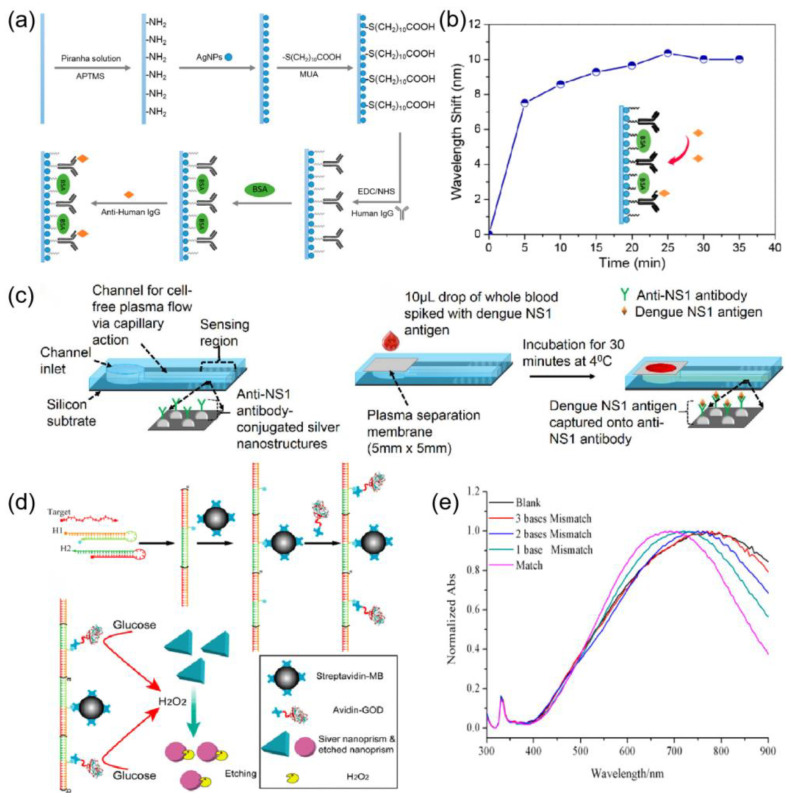
(**a**) Schematic of bio-sensing approaches using the Ag NPs-based LSPR sensor probe. Reprinted with permission from Ref. [127]. Copyright 2015, Sensors. (**b**) LSPR peak wavelength changes during the process of anti-human IgG immobilization on the Ag NP-based sensor surface over a period of 35 min. Reprinted with permission from Ref. [127]. Copyright 2015, Sensors. (**c**) Schematic of the biosensor used with on-chip separation and detection of NS1. Reprinted with permission from Ref. [128]. Copyright 2019, Biosensors and Bioelectronics. (**d**) Construction of HCR-based specific DNA detection platform. The sensing mechanism is based on the etching process of triangular silver nanoprisms. Reprinted with permission from Ref. [129]. Copyright 2014, ACS Nano. (**e**) Specificity of the DNA detection. UV-vis spectra of triangular silver nanoprisms in the presence of target DNA (10 nM) with different mismatched bases. Reprinted with permission from Ref. [129]. Copyright 2014, ACS Nano.

**Figure 12 ijms-24-04142-f012:**
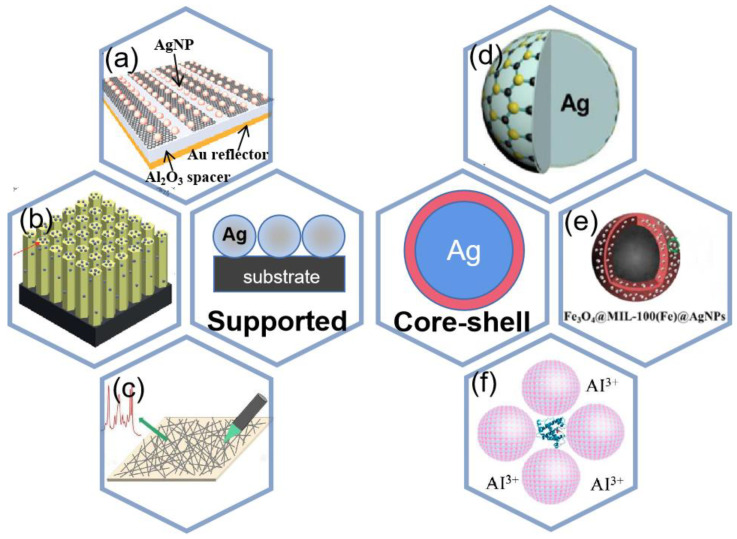
Two different composite modes (**a**) Experimental schematic of the shared substrate. Reprinted with permission from Ref. [138]. Copyright 2021, Small. (**b**) Schematic of the Ag@ZnO chip. Reprinted with permission from Ref. [139]. Copyright 2021, RSC Advances. (**c**) Schematic diagram of silver nanowires array. Reprinted with permission from Ref. [140]. Copyright 2022, Spectrochimica Acta Part A: Molecular and Biomolecular Spectroscopy. (**d**) Schematic of MoS_2_/AgNPs. Reprinted with permission from Ref. [141]. Copyright 2016, Applied Surface Science. (**e**) schematic of Fe_3_O_4_@MIL-100(Fe)@Ag NPs. Reprinted with permission from Ref. [142]. Copyright 2022, Microchemical Journal. (**f**) Schematic Diagram of Experimental Procedure for SERS Detection of Proteins Using Ag IANPs as Substrates. Reprinted with permission from Ref. [143]. Copyright 2020, Analytical chemistry.

**Figure 14 ijms-24-04142-f014:**
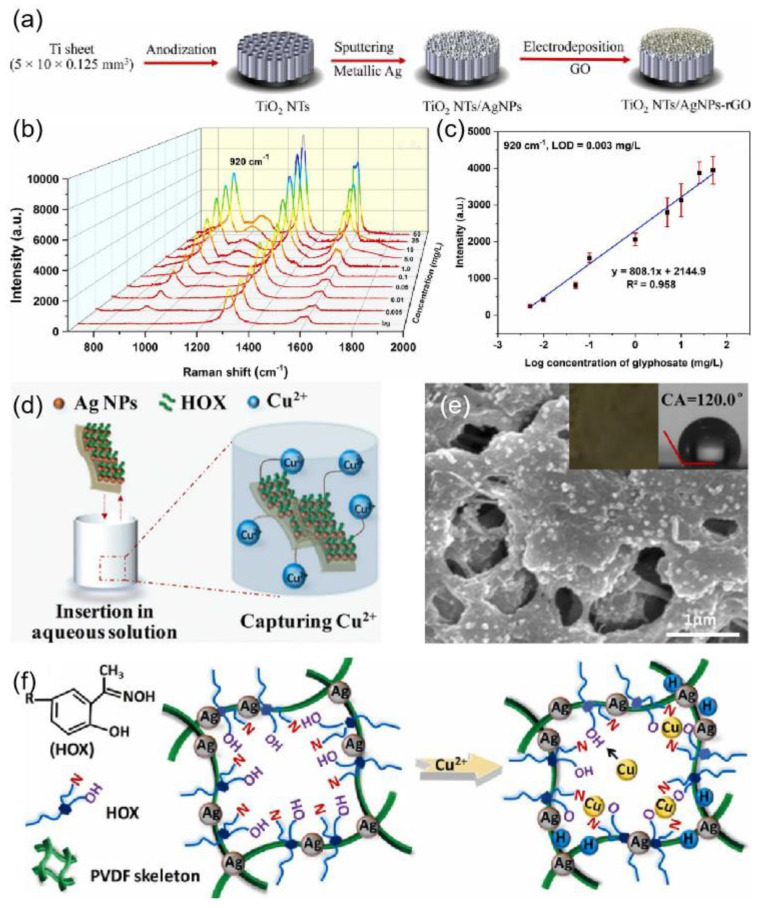
(**a**) Fabrication of the TiO_2_ NTs/AgNPs-rGO hybrid SERS substrate. Reprinted with permission from Ref. [150]. Copyright 2022, Journal of Hazardous Materials. (**b**) Raman spectra of glyphosate with different concentrations on the TiO_2_ NTs/AgNPs-rGO substrate. Reprinted with permission from Ref. [150]. Copyright 2022, Journal of Hazardous Materials. (**c**) Corresponding calibration curve obtained at the characteristic peak of 920 cm^−1^. Reprinted with permission from Ref. [150]. Copyright 2022, Journal of Hazardous Materials. (**d**) Illustration of HOX@Ag-PVDF capturing Cu^2+^. Reprinted with permission from Ref. [151]. Copyright 2022, Journal of Hazardous Materials. (**e**) SEM image of HOX@Ag-PVDF with Cu^2+^. Reprinted with permission from Ref. [151]. Copyright 2022, Journal of Hazardous Materials. (**f**) Capture mechanism of HOX@Ag-PVDF for Cu^2+^. Reprinted with permission from Ref. [151]. Copyright 2022, Journal of Hazardous Materials.

**Figure 15 ijms-24-04142-f015:**
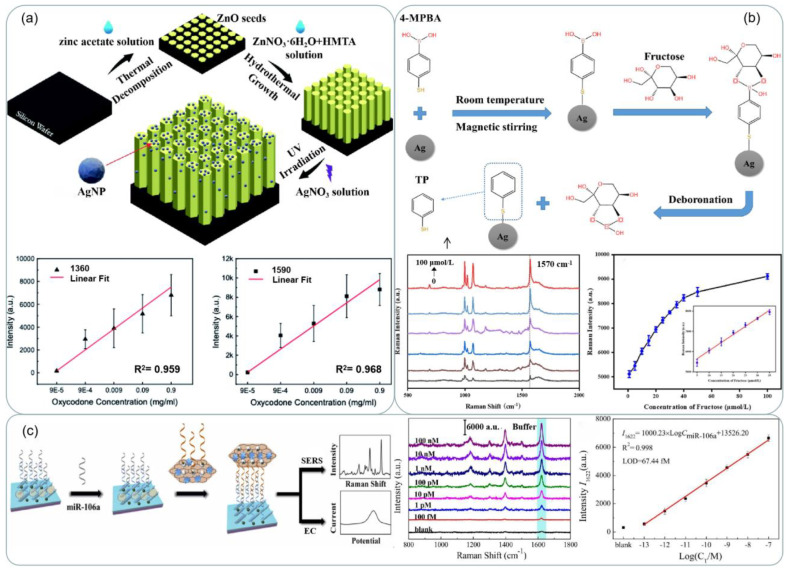
(**a**) Schematic of the Ag@ZnO chip and its fabrication process. The signal intensity versus oxycodone concentrations at the characteristic peaks at 1360 cm^−1^ and 1590 cm^−1^. Reprinted with permission from Ref. [139]. Copyright 2021, RSC Adv. (**b**) Schematic of the synthesis of AgNPs-4MPBA and the process of fructose-induced deboronation reaction. SERS spectra of AgNPs-4MPBA substrate and fructose in urine with concentrations: 0, 0.5, 1, 5, 50, 100 μmol/L, Working curve of fructose concentration and SERS peak intensity at 1570 cm^−1^ in urine. Reprinted with permission from Ref. [153]. Copyright 2023, Spectrochimica Acta Part A: Molecular and Biomolecular Spectroscopy. (**c**) Schematic illustrations of the sandwich-structured sensing system for SERS/electrochemical (EC) dual-mode detection of miR-106a. Reprinted with permission from Ref. [154]. Copyright 2022, Sensors and Actuators B: Chemical.

**Figure 16 ijms-24-04142-f016:**
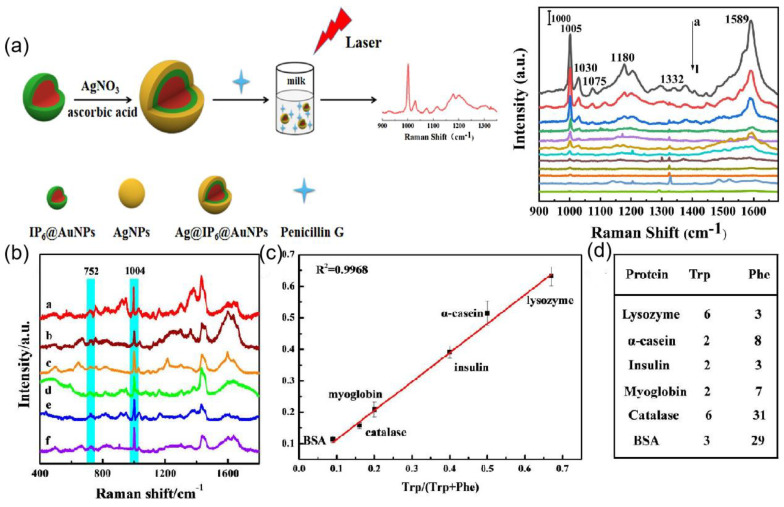
(**a**) Schematic fabrication process of Ag@IP6@AuNPs substrate and the SERS detection of PG. Concentration-dependent SERS spectra of Penicillin G by using Ag@IP6@AuNPs. Reprinted with permission from Ref. [160]. Copyright 2021, Sensors and Actuators B: Chemical. (**b**) SERS spectra of a. lysozyme, b. α-casein, c. insulin, d. myoglobin, e. catalase, and f. BSA at concentrations of 30 μg/mL using Ag IANPs. Reprinted with permission from Ref. [143]. Copyright 2020, Analytical chemistry. (**c**) Correlation curve of I752/I1004 versus Trp/(Trp + Phe). Reprinted with permission from Ref. [143]. Copyright 2020, Analytical chemistry. (**d**) The number of Trp and Phe in six proteins. I752 and I1004 represent the intensity at 752 and 1004 cm^−1^, respectively, and Trp/(Trp + Phe) is the ratio of the number of Trp to the number of Trp + Phe. Reprinted with permission from Ref. [143]. Copyright 2020, Analytical chemistry.

**Table 1 ijms-24-04142-t001:** Primary advantages and disadvantages of different syntheses of silver nanoparticles.

Synthesis Technique	Application	Advantages	Disadvantages	Environment Safety	References
Chemical reduction	Biosensors Catalysis Antimicrobial	Size and structure are controllable Easier to manipulate growth and morphology by simply monitoring the reaction conditions	High cost of the reducing agent Toxic by-products are present	No	[47,48,49]
Green synthesis	Biosensors Catalysis Antimicrobial	Non-toxic chemicals involved Sustainable Cost-effective approach Inherently safer technique	Difficult to control the size and distribution of nanoparticles Mechanism of NP synthesis through bioreduction reaction requires additional experimentation	Yes	[50,51,52]
Seed mediated growth	Bactericidal Catalysis LSPR sensing	High degree of control over the size, shape, and structure	High cost The growth mechanism is not fully understood. Time intensive as well as expensive	Yes	[53,54,55]
Microwave-assisted synthesis	Catalysis Antimicrobial Water purification	Good reproducibility Rapid High yield and improved selectivity Narrow size distribution	In mass production, microwave penetration is insufficient. The principle of microwave-assisted synthesis needs further investigation: thermal and non-thermal effect	Yes	[56,57,58]
Sonochemical mediated synthesis	Catalysis Biomedical	Fast reaction The size of the generated nanostructures is small, and the dispersion of nanoparticles is low	The sonochemical reduction rate depends on the ultrasonic frequency and controlling the shape of the nanoparticles is challenging	Yes	[59,60,61]

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
