# Peer review of "Silver-Based Surface Plasmon Sensors: Fabrication and Applications"

_ijms, 2023, doi:10.3390/ijms24044142_

Round 1
Reviewer 1 Report
The present manuscript reports on the “Silver Based Plasmonic Materials and Applications: A Review”. The work is of some interest but seems to be too primitive and lacks in proper scientific support and justification. Thus, in my opinion, the manuscript in its present form cannot be considered for publication. I recommend major revision.
Following are some of the comments/suggestions which will be useful to the authors.
1. First of all, there are many previous works published on silver nanoparticles. The authors seem deliberately avoid those papers. This is unusual, as the authors need to acknowledge the previous literature demonstrate their research outcomes in terms of advantages and disadvantages. Some of studies are given below need to cited;
i- Begum, R., Farooqi, Z. H., Ahmed, E., Naseem, K., Ashraf, S., Sharif, A., & Rehan, R. (2017). Catalytic reduction of 4‐nitrophenol using silver nanoparticles‐engineered poly (N‐isopropylacrylamide‐co‐acrylamide) hybrid microgels. Applied Organometallic Chemistry, 31(2), e3563.
ii- Arif, M., Farooqi, Z. H., Irfan, A., & Begum, R. (2021). Gold nanoparticles and polymer microgels: Last five years of their happy and successful marriage. Journal of Molecular Liquids, 336, 116270.
iii- Shahid, M., Farooqi, Z. H., Begum, R., Azam, M., Irfan, A., & Farooq, U. (2022). Multi-functional organic–inorganic hydrogel microspheres as efficient catalytic system for reduction of toxic dyes in aqueous medium. Zeitschrift für Physikalische Chemie, 236(1), 87-105.
iv- Shahid, M., Farooqi, Z. H., Begum, R., Arif, M., Irfan, A., & Azam, M. (2020). Extraction of cobalt ions from aqueous solution by microgels for in-situ fabrication of cobalt nanoparticles to degrade toxic dyes: A two fold-environmental application. Chemical Physics Letters, 754, 137645.
v- Arif, M., Shahid, M., Irfan, A., Nisar, J., Wu, W., Farooqi, Z. H., & Begum, R. (2022). Polymer microgels for the stabilization of gold nanoparticles and their application in the catalytic reduction of nitroarenes in aqueous media. RSC advances, 12(9), 5105-5117.
vi- Begum, R., Naseem, K., Ahmed, E., Sharif, A., & Farooqi, Z. H. (2016). Simultaneous catalytic reduction of nitroarenes using silver nanoparticles fabricated in poly (N-isopropylacrylamide-acrylic acid-acrylamide) microgels. Colloids and Surfaces A: Physicochemical and Engineering Aspects, 511, 17-26.
2. The reference in the text is not correct. References are not present in the text section.
3. The equation in line 131 is not correct. Correct it and give a number for this equation.
4. The reaction conditions are not given in the synthetic methods. These conditions are very important for morphology and size of metal nanoparticles.
5. Copywrite is missing for some figures as Figure 17 (a), Figure 14 (a) and (b) etc.
6. Grammatic and topographic mistakes are present in this manuscript.
7. Title of article should be improved.
8. Improve the entire manuscript by a good research specialist.
9. The reasons did not explain by the authors properly.
Author Response
Dear Reviewer,
We have studied the comments carefully, and then revised the whole manuscript. Please see the detail response from attachment file.
Thank you so much!
Qingwei Liao

Reviewer 2 Report
In this article " Silver Based Plasmonic Materials and Applications: A Review", Yinghao Li et al about the Silver Based Plasmonic Materials and their Applications. Overall, based the data provided, it can be considered for publication in this journal. However, there are some issues that have to be fixed before publication;
The title should be revised to somewhat catchy type.
For increasing the interest of readers, the abstract could be more specific in term of scope, objectives and conclusions.
The citation of references show errors please correct.
Please specify the novelty, significance, technical merit of the study in a better way.
English is very poorly presented throughout the manuscript. as well as the presentation is also very poor, should be uniform.
Please revised introduction with proper consequences with some new references such as https://doi.org/10.3390/bios9020078 , Applied Nanoscience 10 (5), 1369-1378,
I am not sure for the permission as the author mentioned data compiled from. please check.
Besides putting the information which were previously published, authors need to write some critical statements on these properly.
Some errors regarding the sub/super script, spacing and typo need to consider throughout the manuscript.
Make sure that the format of references are uniform. Moreover, please add some more references to support this study.
Author Response

(The authors gave the same response as above.)

Reviewer 3 Report
Manuscript ID: ijms-2133381
I have read the manuscript entitled “Silver Based Plasmonic Materials and Applications: A Review”. In this review the authors have described in detail the surface plasmonic response of silver nanostructures and their applications in sensing, photocatalysis, and biomedicine. Overall, the work may be of interest to a broader audience. However, the authors should address the following points outlined below to improve the scientific quality. After the suggested revisions are carefully addressed, this work may be considered for publication in IJMS.
1. Introduction is missing the references. It can be improved by citing more recent work in the field of silver based plasmonic materials.
2. How the current review is better than the previous reported reviews? Explain at the end of introduction.
3. Check the typo and grammatic mistakes like “precuisors” in line 161 also in line 480. Carefully check the whole review.
4. Figures 3a, 4a, 5e-d are not clear, revise them to make clearer.
5. Table containing the comparison of recent synthesis methods and applications should be added in corresponding sections.
6. Some disadvantages of using the silver-based materials in the reported application should also include.
7. The stability study of the silver nanostructures during its application (e.g. in sensing and photocatalysis) must be discussed.
8. I strongly suggest the revision of conclusion with perspective, containing the whole summary.

Author Response

(The authors gave the same response as above.)

Round 2
Reviewer 1 Report
Accept
Reviewer 3 Report
The review article has been revised well and can be accepted now.